# Reference Genes Selection of *Gymnosporangium yamadae* during the Interaction with Apple Leaves

**DOI:** 10.3390/jof8080830

**Published:** 2022-08-09

**Authors:** Chenxi Shao, Wenhao Lao, Yingmei Liang

**Affiliations:** 1The Key Laboratory for Silviculture and Conservation of Ministry of Education, Beijing Forestry University, Beijing 100083, China; 2Museum of Beijing Forestry University, Beijing Forestry University, Beijing 100083, China

**Keywords:** *Gymnosporangium yamadae*, reference genes, pathogenic mechanism, apple

## Abstract

Apple rust disease caused by *Gymnosporangium yamadae* is the one of the major threats to the development of the apple industry in China, but the pathogenic molecular mechanism of the disease remains unclear. It is imperative to screen out appropriate reference genes during the interaction between *G. yamadae* and apple leaves to analyze the gene expression patterns during the pathogenesis of *G. yamadae*. *ACT*, *EF1*, *EF2*, *GAPDH*, *40S*, *60S*, *α-TUB*, *β-TUB* and *UBCE3* were selected as candidate reference genes based on the transcriptomic dataset of *G. yamadae*. The expression levels were tested by real-time quantitative PCR during time-course infection of apple leaves and the expression stabilities were evaluated by *△*Ct method as well as by three software (NormFinder, geNorm and BestKeeper) and one web-based analysis software (RefFinder). The expression stability of the candidate reference genes was further validated by using the effector candidate gene *Cluster-3395.48660* as the target gene in RT-qPCR. According to the results by *△*Ct and BestKeeper, *40S*, *EF2* and *EF1* were the most stable reference genes, while *EF1*, *EF2* and *GAPDH* were the most stable reference genes based on the NormFinder analysis result. The geNorm recommended the most stable genes *EF1*, *EF2* and *α-TUB* as reference genes. Comprehensive analysis results of the RefFinder indicated *EF1*, *EF2* and *α-TUB* were the most suitable genes. Based on these results, *EF1*, *EF2* and *α-TUB* were considered as reference genes for analyzing the gene expression profiles of *Cluster-3395.48660* in different infection stages, and the results were consistent with the transcriptome data. All the results suggest that the combination of *EF1*, *EF2* and *α-TUB* proved to be acceptable reference genes during the interaction between *G. yamadae* and apple leaves.

## 1. Introduction

Apple rust caused by *Gymnosporangium yamadae* leads to apple defoliation and loss of fruit quality, causing serious economic losses in most apple-growing areas in Asia [1,2]. *G. yamadae* is a demicyclic and heteroecious rust fungus that lacks a urediniospore stage, so it is incapable of reinfecting its host [3]. Its aeciospores infect juniper branches and then switch to the telial stage. After overwintering, the teliospores germinate and produce basidiospores that invade apple leaf cells from the upper surface and then, a few days later, orange spermogonia form on the upper surface while hair-like aecia develop in the lower raised surfaces at the same areas of leaves after a period of time [3,4,5]. The aeciospores scatter from the tips of mature aecia and infect the juniper branches [4]. At present, the prevention and control of apple rust mainly depends on the removal of infectious sources and chemical spraying, with limited control effect [6].

Rust genome studies revealed a large repertoire of effector protein genes involved in the pathogenicity mechanisms of rust fungi [7]. These effector proteins are devoted to interfering with defense functions and establishing a compatible interaction with the host, which have high stage, host, and tissue specific transcriptional control during the infection [8,9]. Duplessis et al. [10,11] confirmed the highly dynamic expression profile of *Melampsora larici-populina* candidate effector protein genes during spore germination and the infection stage by transcriptome analysis combined with real-time quantitative PCR (RT-qPCR), which suggested that these candidate effector protein genes might be involved in the complex cellular process of the host during rust pathogen colonization and infection. Qi et al. [12] identified the effector protein gene *PstGSRE1* of *Puccinia striiformis* f. sp. *tritici* was highly up regulated in the early infection stage, which inhibited the reactive oxygen related immune pathways in wheat and, thus, promoted the pathogen infection. However, due to the lack of urediniospores capable of reinfecting the host, inoculation trials using basidiospores can only be carried out under suitable conditions each spring, providing very limited access to obtain spore material for our study to investigate the pathogenic molecular mechanism of *G. yamadae* at the initial stage [13]. Tao et al. [3,14] predicted a number of candidate effector protein genes from transcriptome data of telia, spermogonia and aecia stages of *G. yamadae* that are up-regulated during the infection, including a candidate effector protein gene *Cluster-3395.48660* that was significantly up-regulated in the spermogonia stage that might be associated with pathogenicity. Nevertheless, the biological characteristics of these candidates remain unknown. To further investigate the functional mechanisms of these candidate effectors, the reference genes of *G. yamadae* should be screened out preferentially and used for normalized correction in RT-qPCR to further verify the expression profiles of these candidate effector protein genes in a more accurate and reliable way.

Recently, many studies have screened and identified reference genes in various plant pathogens that are suitable for their specific experimental conditions to analyze the expression of target genes, such as *Hemileia vastatrix* [15], *P. triticina* [16], *P. helianthi* [17], *Uromyces appendiculatus* [18], *Fusarium graminearum* [19] and *Magnaporthe oryzae* [20]. However, these reference genes are not identical. Vieira et al. [15] analyzed the *H. vastatrix* reference genes based on transcriptome data and confirmed that *40S_Rib*, *GADPH*, and *Cyt III* were the most stable in planta gene expression studies. In the interaction between *M. larici-populina* and poplar, *alpha-tubulin* and *elongation factor-1-alpha* were selected as reference genes because of their remarkable expression stability and were then used to analyze gene expression profiles of *M. larici-populina* during its infection stage [21]. Similarly, reference genes such as *α-tubulin*, *β-tubulin*, *β-actin* from *P. striiformis* f. sp. *tritici* and *ubiquitin conjugated enzyme*, *elongation factor 2*, selected from the *P. helianthi* transcriptome data have also been successfully used to analyze gene expression of the rust pathogen [17,22]. Screening studies of *U. appendiculatus* reference genes showed that ten genes, including *actin*, *ATP synthase β subunit*, *cytochrome B*, *elongation factor-1-alpha*, *elongation factor-3*, *glyceraldehyde-3-phosphate dehydrogenase*, *pyruvate dehydrogenase*, *40S ribosomal protein S14*, *tubulin* and *E2 ubiquitin conjugated enzyme* could be analyzed for the standardization of gene expression, while the remaining four genes were discarded as the least stable [18]. Numerous studies have claimed that the stability of reference gene expression varies from species to species, which mainly depends on the tissue type, developmental stage, and experimental conditions, even within the same species [18,23,24,25]. Thus, it is vital to screen reference genes of *G. yamadae* for internal correction of RT-qPCR under specific experimental conditions.

Combined with the above research, the present study chose nine housekeeping genes, including actin (*ACT*), elongation factor 1 (*EF1*), elongation factor 2 (*EF2*), glyceraldehyde-3-phosphate dehydrogenase (*GAPDH*), α-tubulin (*α-TUB*), β-tubulin (*β-TUB*), E3 ubiquitin conjugated enzyme (*UBCE3*), 40S ribosomal protein (*40S*) and 60S ribosomal protein (*60S*) as candidate reference genes from our previous transcriptome datasets of *G. yamadae* teliospores [3]. We employed real-time quantitative PCR to measure the expression of these reference gene candidates at the different infection stages of *G. yamadae* in apple leaves, and then determined the expression stability by using *△*Ct, NormFinder, geNorm, BestKeeper software and RefFinder online software. The selected reference genes were further verified by testing the expression level of candidate effector protein gene *Cluster-3395.48660*. Collectively, three suitable reference gene candidates were selected for the standardization of gene expression studies of *G. yamadae* during infection on apple leaves.

## 2. Materials and Methods

### 2.1. Plant Material and Artificial Inoculation

*Malus domestica* cv. Gala seedlings (3 years old, average 80 cm in height and 1.5 cm in diameter breast high) were planted in flowerpots for inoculation. *G. yamadae* mature telia on juniper were collected from Northwest A&F University, Yangling, Shaanxi. Basidiospores were produced from germinated teliospores and were collected into sterile vials used for inoculation onto apple leaves and stored in a greenhouse, as described [13]. The inoculated apple leaves were harvested from at least three plants at 0, 3, 6, 10, 16, 26, 36, 60, 67 and 77 days post-inoculation (dpi). Healthy apple leaves were used as controls. The leaf samples were flash-frozen in liquid nitrogen and stored at −80 °C until RNA isolation.

### 2.2. RNA Extraction and cDNA Synthesis

Total RNA was extracted from the leaf samples using the RNA Easy Fast Plant Tissue Kit (Tiangen Biotech, Beijing, China) following the manufacturer’s instructions. The integrity of total RNA was verified by 1% agarose gel electrophoresis, and the concentration and purity of isolated RNA was measured with the NanoDrop 2000 spectrophotometer (Thermo Fisher Scientific, Waltham, MA, USA). First-strand cDNA was synthesized from 2 µg RNA of each sample with FastKing RT Kit (With gDNase) (Tiangen Biotech, Beijing, China), according to the manufacturer’s instructions. All synthesized cDNA were stored at −20 °C until use.

### 2.3. Reference Gene Selection, Primer Design and RT-qPCR Parameters

All primers used in RT-qPCR were designed using Primer3 online software (https://bioinfo.ut.ee/primer3-0.4.0/ (accessed on 1 March 2021)) and synthesized by RuiBioTech (Beijing, China). The production size of primers ranges from 90 bp to 200bp and all primers had a GC (composition) content of 40–60% to ensure RT-qPCR amplification efficiency. Nucleotide sequences of *G. yamadae* candidate reference genes and candidate effector genes are presented in Appendix A.

The specificity of the primers was determined by the single peak of the fusion curve in real-time PCR and the single band of the amplification products were detected by agarose gel electrophoresis. PCR products of each candidate reference gene were diluted with 10^6^, 10^7^, 10^8^, 10^9^ and 10^10^ as standard in real-time quantitative PCR for testing the efficiency of these primers. The standard curve, amplification efficiency and correlation coefficient (R^2^) of each primer were directly generated by Bio-Rad CFX Maestro 1.1 software (Bio-Rad Laboratories, Hercules, CA, USA). All RT-qPCR program were performed on a CFX96TM Real Time System (BioRad, Hercules, CA, USA) and by using SuperReal PreMix Plus (SYBR Green) (Tiangen Biotech, Beijing, China). The 20 μL total reaction volume was as follows: 10 μL SuperReal PreMix Plus (2×), 0.5 μL cDNA, 0.6 μL upstream and downstream primers (10 umol/L), RNase-free ddH_2_O supplemented to 20 μL. Each sample was repeated three times with four technical replicates. The reaction procedure used a two-step PCR protocol: an initial denaturation at 95 °C for 15 min, followed by 40 amplification cycles of 10 s at 95 °C and 20 s annealing at 60 °C. A melt curve analysis was started from 60 °C to 95 °C, with each cycle up 0.5 °C after every step. Cq values, melting curves and amplification curves were obtained automatically by Bio-Rad CFX Maestro 1.1 software (Bio-Rad Laboratories, Hercules, CA, USA).

### 2.4. Data Analysis

For determination of the most stable reference gene candidates, *△*Ct method [26] and NormFinder [27], geNorm [28] and BestKeeper [29] software and RefFinder online software (http://150.216.56.64/referencegene.php (accessed on 25 May 2021)) were used according to the software instructions. *△*Ct method and BestKeeper software were based on the original Cq values obtained by RT-qPCR. *△*Ct method was used to compare the relative expression levels of candidate reference genes in each sample and the variation of Cq values among all samples. Meanwhile, Excel was used to calculate the standard deviation (SD) of Cq values of candidate reference genes in each sample to determine the appropriate reference genes [26]. BestKeeper software calculated and compared the standard deviation, coefficient of variation (CV), correlation coefficient (CC [r]) and significance (*p*-value) of Cq values to determine the stability of gene expression, and pair-wise correlation was used to determine the best candidate reference genes [29]. NormFinder and geNorm software were used the data transformed by 2^−^*^△^*^Ct^ of the original Cq values and to calculate the expression stability value (M value) and average expression value M of candidate reference genes, respectively [27,28]. The gene with the best stable performance had the lowest M values. Then the pairwise variation value V_n/n+1_ of candidate reference genes was calculated to determine the optimal number of reference genes needed for normalization [29]. If the cut-off threshold value was V < 0.15, the optimal number of internal reference genes was n [29]. Based on the above results, the original Cq values were comprehensively analyzed by RefFinder and generated a ranking according to the geometric mean of corresponding rankings [30].

For validation of screened candidate reference genes, *G. yamadae* candidate effector protein gene *Cluster-3395.48660* was used as the target gene by performing RT-qPCR to detect the Cq values of the gene at 0, 60 and 77 dpi. The relative expression levels of the gene at different time points were analyzed by Bio-Rad CFX Maestro 1.1 software (Bio-Rad Laboratories, Hercules, CA, USA).

## 3. Results

### 3.1. Primer Specificity and Efficiency Evaluation

All the primers were amplified to obtain fragments of the same size as the target products from *G. yamadae* cDNA. However, melting curves of *ACT*, *60S*, and *β-TUB* had more than one peak when using the healthy apple leaves cDNA as the RT-qPCR template, indicating the presence of non-specific amplification products, so could not be used as reference genes. The remaining six reference gene candidates (*EF1*, *EF2*, *GAPDH*, *α-TUB*, *UBCE3*, *40S*) had a single peak in their primer melting curves, which demonstrated a satisfactory primer specificity. The primer efficiencies of the six candidate reference gene primers were calculated using ten-fold dilution of pooled cDNA and ranged from 89.3% to 114.5% with R^2^ > 0.99 (Table 1).

### 3.2. Expression Levels of Candidate Reference Genes in Different Samples

The Cq values of the six candidate reference genes in all samples were obtained by RT-qPCR. The results showed that the Cq values of all candidate reference genes in the 0 and 3 dpi samples ranged from 23 to 30, while those in the 6 to 77 dpi samples ranged from 17 to 26 (Figure 1A). The Cq values of candidate reference genes increased first, then decreased and stabilized at different infection stages after inoculation (Figure 1A). Comparison of the Cq values of each candidate reference gene in the samples from 6 to 77 dpi, showed that *EF1* had the lowest mean Cq value (18.11) indicating the highest expression level, while *UBCE3* had the highest mean Cq value (22.64) indicating the lowest expression level among the candidate reference genes (Figure 1B). The Cq value variation of *EF2* was 2.56, followed by *EF1* having 2.59, while *UBCE3* had the highest Cq value variation of 4.28 (Figure 1B). The Cq values and their variations of each candidate reference gene were different in all samples, so it was necessary to analyze the expression stability of these candidate reference genes.

### 3.3. Stability of Candidate Reference Genes

We evaluated the expression stability of reference gene candidates in planta at 6 to 77 dpi as their Cq values were similar between different time points after inoculation. Based on *△*Ct and BestKeeper analysis, the stability of these genes was ranked from the most stable to the least stable as *40S* > *EF2* > *EF1*> *α-TUB* > *GAPDH* > *UBCE3* (Figure 2A, Table 2). Then, the M value and average expression stability M were determined by the algorithms NormFinder and geNorm, respectively. The ranking of M value from highest to lowest resulted in: *40S* (0.244) > *α-TUB* (0.207) > *UBCE3* (0.200) > *GAPDH* (0.179) > *EF2* (0.141) > *EF1* (0.036) (Figure 2B). The ranking of average expression stability M from lowest to highest resulted in: *UBCE3* (0.643) > *40S* (0.586) > *GAPDH* (0.536) > *α-TUB* (0.487) > *EF1* = *EF2* (0.386) (Figure 2C). Thus, genes *GAPDH*, *EF1* and *EF2* were estimated to have the highest stability by NormFinder and genes *EF1*, *EF2* and *α-TUB* were estimated to have the highest stability by geNorm. The ranking results of the stability of each candidate reference gene obtained by different methods and software analysis were slightly different, so we conducted a further comprehensive stability analysis by using RefFinder online software. As shown in Figure 2D, the stability rankings were *EF1* > *EF2* > *α-TUB* > *UBCE3* > *40S* > *GAPDH*. geNorm also calculated the pairwise variation V_3/4_ is 0.13, which indicated that the use of these three most stably expressed genes as reference genes of *G. yamadae* would be sufficient (Figure 3).

### 3.4. Validation of Candidate Reference Genes

To validate the selected reference genes *EF1*, *EF2* and *α-TUB*, we conducted a RT-qPCR analysis to detect the expression level of candidate effector gene *Cluster-3395.48660* that was identified in our earlier transcriptome study. *Cluster-3395.48660* had average FPKM of 226 in spermogonia and had average FPKM of 72 in aecia. In RT-qPCR analysis, we used the combination of *EF1*, *EF2* and *α-TUB* to normalize. The relative expression levels of *Cluster-3395.48660* at 60 dpi (the spermogonia stage of *G. yamadae*) and 77 dpi (the aecia stage of *G. yamadae*) were higher than those at 0 dpi. More importantly, the relative expression level at 60 dpi after inoculation was higher than that at 77 dpi (Figure 4). Regardless of which reference gene or combination of reference genes were used for standardization in RT-qPCR, the expression level of *Cluster-3395.48660* was consistent with the result of transcriptome sequencing.

## 4. Discussion

It is a common strategy when performing RNA-seq to screen optimal reference genes in a species under different experimental conditions. Zhao et al. [31] identified different combinations of internal reference genes in Switchgrass suitable for specific experimental conditions based on a previous transcriptome study [32]. Hu et al. [33] successfully screened for the suitable reference genes for different tissues of *Macrobrachium nipponense* based on transcriptome libraries. In the studies of rust pathogens, reference genes were mainly used to analyze the gene expression of rust pathogens when infecting host plants. As most rust pathogens lack genome data, studies on screening reference genes depended on RNA-seq data, such as *P. helianthi* [17], *U. appendiculatus* [18] and *H. vastatrix* [15]. In the present study, candidate reference genes of *G. yamadae* were selected from the transcriptome data in the telia stage that infected junipers.

Generally, reference genes are essential for maintaining basic cellular functions and should be constitutively expressed in various tissues/organs and developmental stages [34,35]. Due to the functionality, genes encoding actin, elongation factor, glyceraldehyde-3-phosphate dehydrogenase, ubiquitin conjugated enzyme and tubulin were commonly used for gene expression analysis by RT-qPCR [15,17,21,36,37]. Ribosomal proteins have also been confirmed to have an excellent stability in some plant pathogens [15,20,38]. Therefore, there is a need to validate the stability of these genes and assess whether they are suitable in gene expression analysis of *G. yamadae*.

In the present study, the timing of sampling was based on the appearance of typical symptoms after the inoculation of *G. yamadae* on apple leaves. The results of the artificial inoculation experiment showed that the course of apple rust was relatively regular and stable, taking into account the experience gained from our previous research and the climatic conditions of the year [13,14,39]. The course of the apple rust can be divided into the following stages: the initial stage of basidiospores invasion from 0 to 3 days without macroscopic symptoms on apple leaves; the colonization stage from 6 dpi with many chlorotic spots appearing on the leaves, indicating the formation of haustorium [13]; the propagation and infection stage from 10 dpi to 77 dpi including the formation and development of rust spermogonia and aecia. After colonization, *G. yamadae* obtained nutrients from the host leaves for propagation and development. During the process, the rust pathogen expanded in the host cells, accompanied by the increase of biomass accordingly. Considering the above characteristics of the developmental phase of *G. yamadae*, 2 μg total RNA was added in a reverse transcription experiment in the preparation of template cDNA for RT-qPCR to ensure that the same amount of cDNA was present in each sample. However, there was only a fraction of the rust RNA in the total RNA in the in-planta samples at the initial stage of the rust invasion. Therefore, it was inapposite to take the samples of 0 and 3 dpi into consideration for further analysis of the variation of Cq value and expression stability of candidate reference genes.

The primary criterion for screening reference genes is the expression abundance of reference genes, which is inversely proportional to the Cq value [40]. Generally, the Cq value should be between 15 and 30. In this study, the mean Cq values of all selected candidate reference genes were higher at both 0 and 3 dpi than at other infection stages, whereafter the Cq value decreased with the development of rust disease (Figure 1A). The results indicated that the abundance of candidate reference genes was lower in samples of 0 and 3 dpi and higher in samples from 6 to 77 dpi, which were consistent with the macroscopic symptoms of the increased rust biomass after colonization and expansion in the host leaves [15,17,21]. In addition, the variation between Cq values of each reference gene under different conditions is also an important criterion for screening reference genes [26]. In our data, *EF2* had the smallest variation of Cq value, while *UBCE3* had the largest variation of Cq value among candidate reference genes, which preliminarily suggested that *EF2* had the best stability, while *UBCE3* had the worst.

To guarantee the accuracy and reliability of the results of internal reference gene stability analysis, the *△*Ct method and the BestKeeper, NormFinder and geNorm software commonly used for screening reference genes were selected to analyze the experimental data, and RefFinder was used to comprehensively evaluate the results [20,30,41]. The analysis were necessary to select suitable reference genes with minimal variability and high stability in *G. yamadae*. However, there were some differences in the results based on different methods and software. Based on the *△*Ct method and BestKeeper software, the candidate reference genes with the best stability were *40S*, *EF2* and *EF1*. NormFinder showed that *GAPDH*, *EF2* and *EF1* had the best stability, while geNorm considered *EF1* and *EF2* to be best and *α-TUB* the third best. These different evaluation results were due to the *△*Ct method and BestKeeper directly analyzing the original Cq value, while NormFinder and geNorm use different algorithms to analyze the original Cq value after 2^-^*^△^*^Ct^ transformation [27,28]. None of these approaches can select a fully satisfying gene as reference gene, so we performed a comprehensive assessment of reference genes based on the results derived from the algorithms employed. According to comprehensive analysis of RefFinder, we identified *EF1*, *EF2* and *α-TUB* were reference genes of *G. yamadae*. Although the pairwise variation values of reference genes obtained by geNorm were all less than 0.15 for V_3+4_ and V_4/5_, this indicated that 4 or 5 genes could be used as reference genes. Based on the stability of candidate reference genes in this study, there was no need to use more than three reference genes for normalization in the gene expression analysis [31,32].

To validate the selected reference genes, we chose the candidate effector protein gene *Cluster-3395.48660* predicted in the transcriptome of *G. yamadae* as the target gene for RT-qPCR in this study [14]. In the transcriptome data, candidate effector gene *Cluster-3395.48660* identified in spermogonia has FPKM of 226, and the transcript in aecia (gene ID is Cluster-17511.35872) has FPKM of 72 (Appendix A). The expression level of *Cluster-3395.48660* was detected using a combination of *EF1*, *EF2* and *α-TUB* reference genes for normalization. The RT-qPCR data showed that *Cluster-3395.48660* was up-regulated in both 60 and 77 dpi (infection stages) compared to 0 dpi. More importantly, the relative expression level was higher in the spermogonia stage than that in the aecial stage. Consistent with other research that that utilized two or more reference genes, RT-qPCR is more profound and reliable [15,34,42,43]. Collectively, the combination of *EF1*, *EF2* and *α-TUB* can play a role in normalization correction for gene expression analysis by RT-qPCR. In addition, the results also suggested that the candidate effector protein Cluster-3395.48660 might play a role during the infection of *G. yamadae* on apple leaves, which should be a focus in further research.

## 5. Conclusions

Here, we systematically screened and identified the stability of nine candidate reference genes of *G. yamadae* in different infection stages for the first time. *EF1*, *EF2* and *α-TUB* were finally identified based on *△*Ct method, NormFinder, geNorm, BestKeeper and RefFinder software evaluation. The candidate effector protein gene *Cluster-3395.48660* was used to validate *EF1*, *EF2* and *α-TUB*, which indicated that the combination of *EF1*, *EF2* and *α-TUB* could be suitable as endogenous control genes for further gene expression studies during the infection of *G. yamadae* on apple leaves.

## Figures and Tables

**Figure 1 jof-08-00830-f001:**
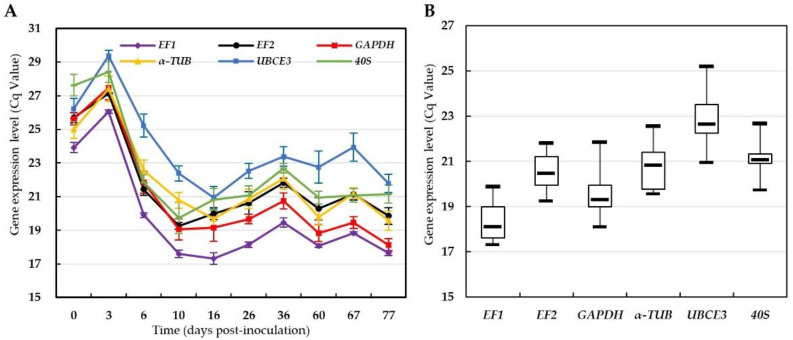
Cq values for the candidate reference genes. (**A**) Plot of Cq values of the candidate reference genes in the test infection stages; (**B**) Box plots of the Cq values of the candidate reference genes from the samples except 0 dpi and 3 dpi. The boxes depict the quartiles, the bars in the box plots denote the minimum and maximum values, and the line in the box indicates the median value.

**Figure 2 jof-08-00830-f002:**
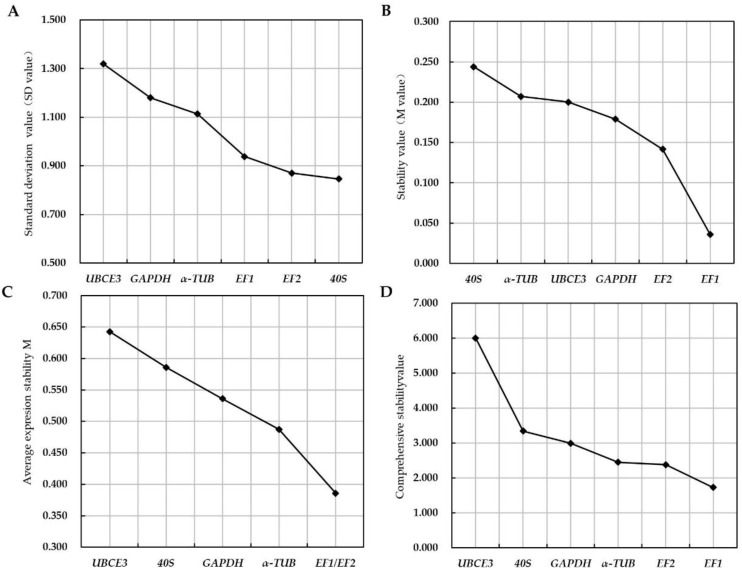
The stability analysis of candidate reference genes. (**A**): Ranking by the *△*Ct method; the graph shows the standard deviation of average expression value for the candidate reference genes; (**B**): Ranking by the NormFinder algorithm; the graph shows the stability value for the candidate reference genes; (**C**): Ranking by the geNorm algorithm; the graph shows the stability value for the candidate reference genes; (**D**): Ranking by the RefFinder; the graph shows the comprehensive stability values for the candidate reference genes. Genes represented on the abscissa from left to right means the least stable to the most stable.

**Figure 3 jof-08-00830-f003:**
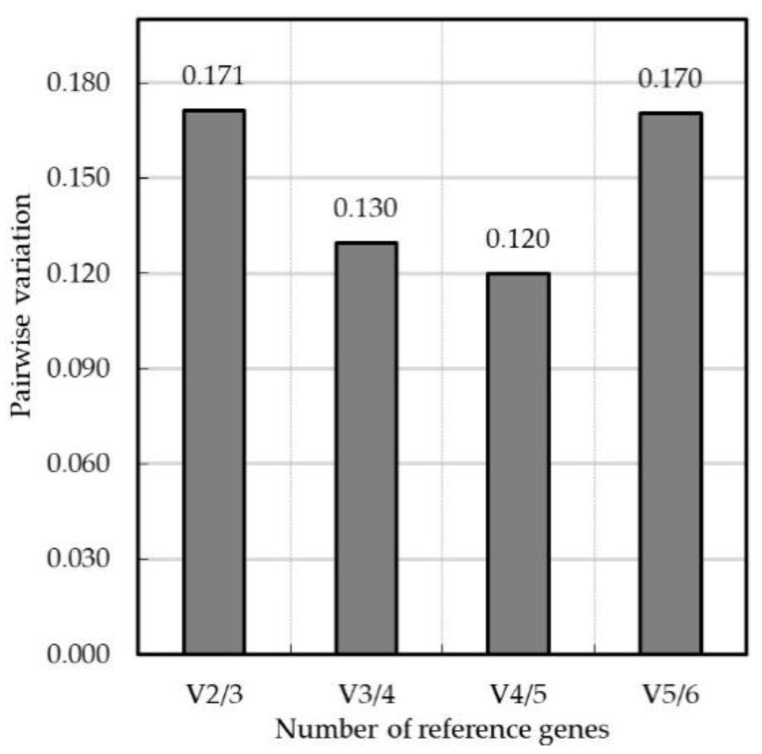
Determination of the number of reference genes by geNorm. The value of V_3/4_ is smaller than the threshold 0.15 shows the most appropriate number of internal factors is 3.

**Figure 4 jof-08-00830-f004:**
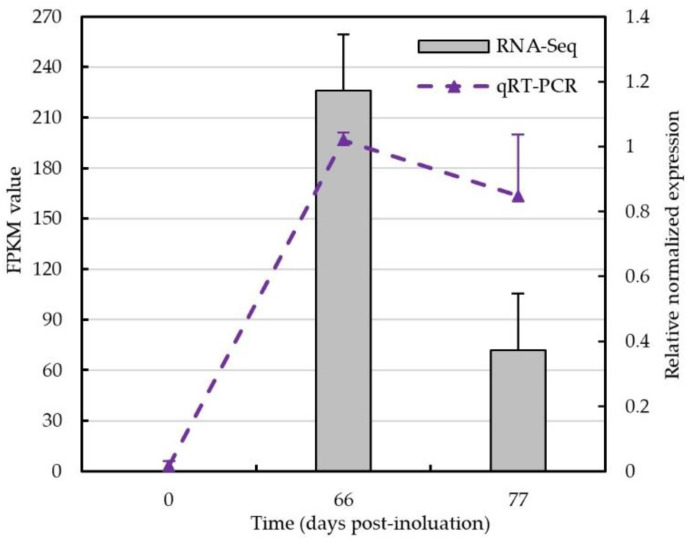
The expression level of the candidate effector gene *Cluster-3395.48660*. The expression level of the candidate effector gene *Cluster-3395.48660* was test in 0 dpi, 60 dpi and 77 dpi planta samples by qRT-PCR. The combination of *EF1*, *EF2* and *α-TUB* were used for normalization. Bars represented FPKM of the RNA-Seq data; dotted lines represent relative expression levels obtained by qRT-PCR; error bars indicate standard error.

**Table 1 jof-08-00830-t001:** Primers for real-time quantitative PCR.

Gene	Description	KOG ID	Sequence (5′-3′)	Amplicon Size (bp)	Efficiency (%)	R^2^
*ACT ^a^*	Actin and related proteins	KOG0676	GACCATGTACTCGGGCATCTCAGCGAAGCCAGAATAGACC	139	-	-
*EF1*	Translation elongation factor EF-1 alpha	KOG0052	AGGAGGCTCAATAGCGTCAA	97	89.30	0.9970
			CAACATGCAATGGTTCAAGG			
*EF2*	Elongation factor 2	KOG0469	AGAAGCGAGGTCACGTGTTT	155	98.40	0.9960
			GTGGTCGAACACCATCTGTG			
*GAPDH*	Glyceraldehyde 3-phosphate dehydrogenase	KOG0657	TCCTGCCTTTGAAATTTTGG	101	114.50	0.9910
			TTGCTTTACGCTTGATGTGC			
*40S*	40S ribosomal protein S7	KOG3320	GGATCCATCGGTGTGAAATC	104	91.16	0.9976
			CTCGGTCACGAACACTCAAA			
*60S ^a^*	60S ribosomal protein L14/L17/L23	KOG0901	ATAGACGCGTTTTCGCTTGT	195	-	-
			CGGTAAAAGGCTCGTCTCAG			
*β-TUB ^a^*	Beta tubulin	KOG1668	CCGATCAATTCACAGCAATG	195	-	-
			CAGGGGGTACCTCATCTTCA			
*α-TUB*	Alpha tubulin	KOG1376	ATATTTCCCGGAGCCAGTCT	106	95.50	0.9950
			TACCATCGAGCATGGTTTGA			
*UBCE3*	E3 ubiquitin-protein ligase	KOG0939	CCTTTGCTGGACTTTGAAGC	183	103.10	0.9910
			GCCTACACCAGAGGAGCTTG			
*Cluster-3395.48660 ^b^*	-	-	GGAAGTGGCACCAACAAAGT	84	-	-
			ATCCACGACATTCGCATACA			

^a^: The primers that have nonspecific amplification in the healthy apple leaves sample (CK), which were abandoned detection of amplification efficiency. ^b^: The candidate effector gene that selected as target genes for the reference genes validation in real-time quantitative PCR.

**Table 2 jof-08-00830-t002:** Expression stability of the candidate reference genes evaluated by BestKeeper.

Parameters	Gene					
	*EF1*	*EF2*	*GAPDH*	*α-TUB*	*UBCE3*	*40S*
Geo Mean	18.35	20.53	19.58	20.79	22.83	21.15
AR Mean	18.37	20.55	19.61	20.81	22.87	21.17
Min	17.31	19.25	18.11	19.57	20.94	19.74
Max	19.90	21.81	21.85	22.56	25.22	22.69
SD	0.77	0.71	0.86	0.86	0.98	0.55
CV (%)	4.21	3.43	4.38	4.12	4.30	2.61
x-fold Min [x-fold]	−2.04	−2.43	−2.76	−2.33	−3.71	−2.66
x-fold Max [x-fold]	2.94	2.43	4.85	3.43	5.22	2.89
x-fold SD [± x-fold]	1.71	1.63	1.81	1.81	1.98	1.47
CC [r]	0.991	0.915	0.930	0.938	0.892	0.768
*p*-value	0.001	0.001	0.001	0.001	0.003	0.026

## Data Availability

Data is contained within the article or Appendix A.

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
