# Peer review of "Reference Genes Selection of Gymnosporangium yamadae during the Interaction with Apple Leaves"

_jof, 2022, doi:10.3390/jof8080830_

Round 1
Reviewer 1 Report
Dear authors,
The paper is very interesting for the region studied as well as for the entire scientific community. Please process the paper technically based on the instructions for authors requested by the journal.
Best regards
Author Response
Thank you for your time reviewing our manuscript.
Reviewer 2 Report
Comment 1: Lines 43-68, Rewrite second paragraph of the introduction is far away from the objective of the study. It will be better to focus on genomic background of the disease progression specific to targeted fungal pathogen and how the putative candidates involve in this plant-fungal interaction.
Comment 2: Line 218, What does mean by “Best genes right”?
Comment 3: Data shown in figure 3 is not clear to me.
Comment 4: Authors have mentioned 6 dpi in line 111 while same is denoted by 6.5 in figure 1A.
Comment 5: lines 238-240, 60 dpi and 77 dpi are too broad range for expression analysis. Authors also need to consider either or both 6 dpi and 10 dpi as these stages recorded ideal Cq values (Figure1A) and will also represent all the major stages of infection as mentioned in lines 273-277. It will be easier to follow the expression pattern if authors also provide expression data from transcriptome.
Comments 6: Considering expression pattern and other considered parameters, 40s also looks like a stable reference gene as it has least fluctuation in Cq values after 6 dpi, therefore it should also be validated by estimating the expression pattern of a putative candidate gene.
Comment 7: Validating the expression of a single target gene is not enough to justify the efficiency of selected reference genes. Authors need to consider few additional targets selected based on different biological backgrounds.
Some minor corrections/ comments
Comment 1: Remove the space between “Similarly, reference” in line 79.
Reviewer 3 Report
Interesting manuscript. I recommend acceptance.
Author Response

(The authors gave the same response as above.)

Round 2
Reviewer 2 Report
I am satisfied with the responses/corrections made by authors against raised comments. Therefore, I would like to recommend this manuscript for publication in this journal.